# Effects of Physical Exercise on Emotional Intelligence from Birth to Adolescence: A Systematic Review Protocol

**DOI:** 10.3390/healthcare12232437

**Published:** 2024-12-04

**Authors:** Falonn Contreras-Osorio, Enrique Cerda-Vega, Christian Campos-Jara, Rodrigo Ramirez-Campillo, Nuria Pérez-Romero

**Affiliations:** 1Exercise and Rehabilitation Sciences Institute, Faculty of Rehabilitation Sciences, Universidad Andres Bello, Santiago 7591538, Chile; falonn.contreras@unab.cl (F.C.-O.); enrique.cerda@unab.cl (E.C.-V.); rodrigo.ramirez@unab.cl (R.R.-C.); 2Exercise and Rehabilitation Sciences Institute, Postgraduate, Faculty of Rehabilitation Sciences, Universidad Andres Bello, Santiago 7591538, Chile

**Keywords:** emotion, social cognition, sport, adolescence, emotion regulation

## Abstract

Background: Emotional intelligence (EI) can be understood as a set of traits or abilities that may have an impact on academic, professional, or mental health. The aim of this protocol was to establish methodological guidelines for a systematic review and meta-analysis of the effects of acute and chronic physical exercise on EI from birth to 21 years (late adolescence). Methods: This protocol followed PRISMA-P guidelines and will be modified in PROSPERO after peer review. The review will include experimental randomized and non-randomized control studies involving physical exercise interventions. PubMed, Web of Science, Scopus, and EBSCO will be utilized for study selection following the publication of the protocol. The risk of bias will be assessed using the ROBINS-I tool, ROB-2, and the GRADE approach will evaluate the certainty of evidence. Random effect meta-analyses will analyse the effect of physical exercise compared with control groups, using effect sizes measures (Hedges’ *g*), with a 95% confidence interval and prediction interval, for each EI outcome (perception, facilitation, understanding, regulation, and management of emotions). Potential moderators, such as exercise intensity, duration, and sociocultural factors, will be analysed. Heterogeneity will be assessed using the I^2^ statistic, and sensitivity analyses will be performed to ensure result robustness. Subgroup analyses may be conducted based on age groups and specific EI dimensions. Conclusions: Publication bias will be assessed using Egger’s test and the trim-and-fill method. The future results aim to provide a foundation for analysing the impact of physical exercise on EI development, potentially guiding future interventions in health, educational, and related fields.

## 1. Introduction

The nature of emotional intelligence (EI) has been widely debated, with theorists such as Petrides and Furnham [1] understanding it as emotional traits and Salovey and Mayer [2] seeing it as a set of abilities. Within the ability models, Salovey and Mayer’s 1997 model [3] and its updates [4] conceive EI through four hierarchical dimensions: (i) perception: identifying and recognizing emotions; (ii) facilitation: using emotions to facilitate cognitive processes; (iii) understanding: correctly interpreting and analysing emotions, their causes, and consequences; and (iv) management: regulating and modifying one’s own emotions and those of others in a conscious and deliberate manner. Moreover, these abilities are described both intrapersonally and interpersonally. According to Mortillaro and Schlegel in 2023 [5], management aligns with the concept of regulation in the intrapersonal context (emotional regulation), whereas it is referred to as emotion management in the interpersonal context.

However, emotional aspects have not only been addressed from the EI point of view [1,2] but also from different theoretical perspectives such as emotional regulation [6] or social cognition [7]. In response to this diversity, there has been a growing need to enhance the measurement of emotional intelligence by connecting its key dimensions with other models in the emotional literature [5]. Thus, the dimensions of EI are linked to other theoretical conceptions, which are often separated and assessed with different instruments in the literature. There are also connections between the dimensions of the EI ability model and other emotional models. People with high EI effectively manage both positive and negative situations by using diverse strategies, seeking social support, and fully experiencing their emotions while being able to regulate them effectively in response to environmental demands [8]. Therefore, they experience less mood impairment, lower emotional intensity, and reduced cortisol secretion in the face of stress, facilitating their adaptation [9,10]. From childhood to young adulthood, enhancing emotional abilities can be essential for personality development, social relationships, and problem solving [11], contributing to the formation of emotionally skilled adults who experience multiple benefits either academically, professionally, or in their mental health [12,13].

Interventions that have focused on improving EI in children and adolescence have been developed through different theoretical perspectives [14]. According to the meta-analysis realized by Hodzic et al. in 2018 [15], among the most used strategies are lectures, role-playing, group discussions, pair work, and reading. Among the initiatives that use physical activity as an intervention strategy, the study by Luis-de Cos et al. [16], which integrated physical exercise with emotional aspects through an emotional physical education program for 14 weeks, including specific activities to promote the identification and regulation of emotions through body expression, dramatization, cooperative workshops, relaxation techniques, yoga, and competitive and cooperative games, stands out. Pre-test and post-test analyses showed a significant increase in the subjective perception of emotional clarity assessed with the Trait Meta Mood Scale (TMMS-24) in the experimental group with respect to the control, with a small effect size (*d* = 0.116). Other authors [17], although theoretically, have planned the integration of EI training through physical exercise using the tripartite model, proposing activities focused on the five main dimensions of EI (i.e., identifying, expressing, understanding, regulating, and using emotions) through sport and physical activity. Because physical-sport activities are based on social interaction among their participants [11,18], they could be the ideal environment for the development of emotional abilities.

Previous systematic reviews have explored the relationship between physical exercise and emotional aspects [11,19,20]. In 2023, D’Cruz et al. [19] analysed the association between the level of physical activity and emotional self-regulation in children up to 7 years of age, including 47 studies in total. This review found a positive association between physical activity and self-regulation in early childhood in 17 of the included studies, particularly in behavioural self-regulation, although the authors described their overall results as inconsistent. On the other hand, in 2019, Ubago-Jiménez et al. [20] conducted a systematic review that included a total of 24 selected studies, analysing the relationship between physical sport activity and perceived EI in students and athletes of various ages (from elementary school to over 40 years old). This last review revealed that, in most of the included studies, the practice of physical activity was related to high levels of EI. Although these findings are relevant for understanding the relationship between both concepts, there is a lack of longitudinal experimental studies and meta-analyses that quantify the effect size of interventions based on physical activity, exercise, or sport on outcome measures that address EI from all its dimensions. In this regard, a systematic review (24 included studies) reported improvements in positive emotions in children and adolescents following physical activity interventions (standard mean difference [SMD] = 0.62; *p* < 0.01), particularly in participants over 12 years old (SMD = 0.73; *p* = 0.01) and following interventions of 30–60 min (SMD = 1.07; *p* < 0.05), showing a more pronounced effect than interventions less than 30 min (SMD = 0.30) and >60 min (SMD = 0.25) [11].

The aforementioned reviews have analysed particular emotional aspects, such as self-regulation or positive emotions, from their respective theoretical perspectives. However, the future systematic review proposes a broader and more integrative view to analyse the effects of physical exercise on the dimensions of EI: perception, facilitation, understanding, regulation, and emotional management, from birth to adolescence, considering the different theoretical perspective that Mortillaro and Schegle [5] propose for each dimension of EI. In addition, this systematic review proposes analysing the moderating effect of variables related to EI, physical exercise (e.g., intensity, duration of intervention, weekly volume), sociocultural aspects, and the type of study. Unlike previous studies, which have not integrated these factors, this research could offer more precise results adapted to different realities in both clinical and educational contexts. Therefore, the aim of this protocol was to establish methodological guidelines for a systematic review and meta-analysis of the effects of acute and chronic physical exercise on EI—perception, facilitation, knowledge, regulation, and emotional management—from birth to 21 years (late adolescence). Despite evidence on the psychological benefits of exercise, the relationship between physical activity and emotional intelligence (EI) remains underexplored. Given that EI is key to stress management and social interactions, understanding how exercise influences EI is individually and socially relevant. This research could provide a basis for interventions that strengthen EI through exercise, benefiting health professionals and the wider community.

## 2. Materials and Methods

This protocol was structured based on the Preferred Reporting Items for Systematic reviews and Meta-Analyses guidelines for protocols (PRISMA-P [21]), and the methodology described is similar to that used in previous studies [22,23]. The protocol was registered in the PROSPERO platform with the following number: CRD42024584195. The registration data will be modified in PROSPERO after publication of the protocol if the peer reviewers consider any improvement necessary.

### 2.1. Eligibility Criteria

The Population, Intervention, Comparison, Outcome, and Study design (PICOS) strategy will be used to define the inclusion and exclusion criteria (Table 1). For the population, studies that recruited participants between 0 (neonates) and 21 years of age (late adolescence [24]) will be included. This criterion was established because of the behavioural, emotional, and brain development characteristics [25,26] that could influence the results obtained. Regarding the intervention, any type of acute or chronic intervention (≥8 weeks) will be considered [27]. In the case of multiple measurements, priority will be given to using the pre- and last measurement, excluding intermediate ones.

### 2.2. Search Strategy

The search will be conducted through the following databases: PubMed, Web of Science (Core Collection), Scopus, and EBSCO. Studies that meet the eligibility criteria found through other sources (e.g., references of previous review studies found in the databases) will also be incorporated. A pilot search was conducted in PubMed on 28 October 2024, yielding 2106 results. The final search will be reflected in a flow chart.

Two researchers (N.P.-R. and F.C.O.) will perform the systematic search without limitations based on participant gender, language, or publication year. Article searches in various databases will commence after the protocol has been approved for publication. The PICOS search approach will be employed for this purpose, utilizing medical subject headings (MeSHs) and free-text keywords (Table 2).

### 2.3. Selection Process

The studies found through the different databases will be uploaded to the Rayyan.ai (https://www.rayyan.ai/ accessed on 29 November 2024). This platform allows its users to carry out the selection process of the systematic review articles. First, the articles with the greatest similarity will be detected, and subsequently, an author (N.P.-R.) will review each of the similarities, eliminating the articles that are really duplicated. After this, the titles and abstracts will be read, where two authors in parallel and blind will evaluate their eligibility according to the established criteria (N.P.-R. and F.C.O.) through the same software mentioned. The reference lists of the articles and reviews identified during the search will also be assessed to select potentially eligible studies. If there are any disagreements among the authors, these will be addressed through consensus with a third author (E.C.-V.). The selection of the studies that make up the review will be recorded by means of a flow chart, as established in the PRISMA guidelines [28], detailing the selection process and the corresponding reasons for exclusion. These data will also be recorded through an Excel document that allows Rayyan.ai to be exported, recording each decision taken.

### 2.4. Data Extraction and Management

For each of the included studies, the following data will be identified: author(s) and year of publication, country of origin, study design, sampling methods, sample size, funding information, participant characteristics (sex, mean age and standard deviation of age, fitness level, comorbidities, treatments and social information (e.g., level of social support, family structure, academic level, socioeconomic level)), description of the physical exercise program (total duration in weeks, follow-up time, weekly frequency, duration of sessions in minutes and intensity of sessions and/or exercises (e.g., heart rate or Borg scale)), and description of the control condition.

The dimensions of EI assessed (e.g., perception, facilitation, comprehension, regulation, management) will also be described, as well as the level from which it is assessed (knowledge, abilities and traits) according to the theory to which the author ascribes, in addition to the tasks or tests used for the assessment (e.g., Mayer–Salovey–Caruso Emotional Intelligence Test ([MSCEIT] [29]).

The results are expected to be analysed, according to availability, in terms of (i) EI as a general index, (ii) each of the abilities within EI (perception, facilitation, knowledge, regulation, and emotional management) and (iii) each of the levels (knowledge, abilities, and traits).

The mean values and standard deviations for each outcome will be recorded, taking into account the period before and after the intervention. Data extraction will be conducted by one author (N.P.-R.) and verified by a second author (F.C.O.). Each author will independently store the data in a Microsoft Excel spreadsheet (Microsoft Corporation, Redmond, WA, USA). Additionally, intercoder reliability will be assessed by piloting ten randomly selected articles and refining the process as needed [30]. Once the data extraction sheet is finalized, one reviewer (N.P.-R.) will perform the initial extraction for all included articles, and a second reviewer will check the entire process (F.C.-O.). In cases where the published data are insufficient, additional information will be requested from the corresponding authors. Any conflicts will be resolved through consensus with a third author (E.C.-V.). The final data will be presented in tables.

If the data are not clear or complete in the paper, procedures for contacting the authors will be applied [31]. In this regard, the authors of these studies will be contacted up to a maximum of two times within two weeks. In case of a lack of response or lack of necessary data, the study will be excluded from analysis. In the case of discrepancies, these will be resolved by consensus with a third author (R.R.-C.).

### 2.5. Risk of Bias in Individual Studies

The risk of bias will be assessed using the Risk of Bias 2 (ROB-2) tool for randomized studies [32,33] by classifying them as “Low risk”, “High risk”, and “Some concerns” of bias according to the analysis of five domains and one overall domain: bias derived from the randomization process, bias due to deviations from the intended interventions, bias due to lack of outcome data, bias in outcome measurement, bias in the selection of the reported outcome, and overall bias. On the other hand, for non-randomized studies, Risk of Bias In Non-randomized Studies of Interventions (ROBINS) [34], will be used, classifying them as “Low Risk”, “Moderate Risk”, “Severe Risk” and “Critical Risk” of bias according to the analysis of seven domains and one overall domain: confounding bias, bias in the selection of study participants, bias in the classification of interventions, bias due to deviations from intended interventions, bias due to missing data, bias in the measurement of outcomes, bias in the selection of the reported outcome, and overall bias. For the assessment of overall bias, in each study, the domain with the highest risk of bias will be identified, and that risk will be used as a reference for the overall assessment of the study [35].

Two authors (N.P.-R. and F.C.-O.) will assess the risk of bias independently, resolving discrepancies with a third author (E.C.-V.). This evaluation will be shown by means of two graphs, one for each item of the tool and another showing the evaluation of each article. In addition, the GRADE (Grading of Recommendations, Assessment, Development, and Evaluation) method will be applied to synthesize and evaluate the certainty of the evidence for each result, with categorizations of very low, low, moderate, and high [36,37].

Regardless of the results derived from the risk of bias and certainty of evidence, all studies that meet the inclusion criteria will be considered for the analysis, considering such information for the interpretation (e.g., moderator analysis according to RoB) and discussion of the results.

### 2.6. Meta-Analysis

When the data have been extracted, it will be determined whether it is possible to perform a random-effect meta-analysis on a given outcome, having at least 3 outcome measures for each outcome, from different studies. Alternatively, if only 2 studies are available, but these studies include an n > 800, their analysis will be considered. The Hedges’ effect size (ES) will be calculated with a 95% confidence interval (CI) and 95% prediction interval. The ES will be derived using the mean and standard deviation values from the experimental and control groups, before and after intervention period. If studies provide data in formats other than mean and/or standard deviation, appropriate statistical transformations will be applied prior to conducting the meta-analysis. The post-intervention standard deviation will be used for data standardization. The meta-analysis will utilize the DerSimonian and Laird method.

The estimated ES will be evaluated using the following categorizations: <0.2 trivial, 0.2–0.6 small, >0.6–1.2 moderate, >1.2–2.0 large, and >2.0 very large [38]. It should be noted that an attempt will be made to identify outlier ES values. An outlier is an extreme case that deviates significantly from the rest of the data and may impact the validity and robustness of a meta-analysis or meta-regression [39,40], as well as the interpretations, conclusions, and inferences derived from it. Although there is no single method for identifying an outlier [39], in exercise intervention studies, ES values ≥ 3.0 (i.e., an improvement of ≥3 standard deviations from the mean) are unlikely after most interventions and will therefore be considered outliers [39].

In research with multiple intervention groups and a single control group, the control group’s sample size is proportionally allocated to enable comparisons across groups. The I^2^ statistic will evaluate heterogeneity, with thresholds of <25%, 25–75%, and >75% representing low, moderate, and high levels, respectively [41].

To evaluate the potential for publication bias in continuous variables (when there are at least 10 studies per outcome), Egger’s test will be applied [42,43]. If publication bias is detected, a sensitivity analysis will be conducted using the trim-and-fill method [44], with the L0 estimator serving as the default to estimate the number of missing studies [45]. A multivariate meta-regression using the DerSimonian and Laird random-effects model will be conducted to determine whether continuous moderators (e.g., training frequency, duration, or the total number of intervention sessions) influence the effects of the interventions on the dependent variables. A calculation of meta-regression will be performed with at least 10 studies per covariate [46]. In addition, a sensitivity analysis will be performed (study by study) to assess the robustness of the summary estimates (e.g., *p*-value, ES, I^2^). All analyses will be performed with Comprehensive Meta-Analysis software (version 4, Biostat, Englewood, NJ, USA). Statistical significance will be set at *p* ≤ 0.05.

### 2.7. Moderators

In the present proposal, some potential moderators will consider the effects of physical exercise on measures of EI, based on previous studies [15,47]. Regarding EI variables, the approach (ability, trait, and mixed models) and theoretical models will be considered as moderators. Regarding exercise, intensity, duration of the intervention (weeks), weekly training volume (minutes), type and modality (according to the activity, place of realization, and number of people involved), and previous experience level will be analysed. Social moderators such as level of social support, family structure, or academic level will also be considered. Finally, the type of study (randomized or non-randomized) and age will be considered.

When applicable, analyses will utilize the median splitting technique. The median will be determined only if data for a specific moderator (across its categories) are available from at least three studies. If a study includes two experimental groups with identical information for a given moderator, only one group will be included to prevent overrepresentation in the median calculation. Additionally, to reduce heterogeneity, medians will be calculated exclusively from studies reporting data for the specific outcome being analysed, rather than applying an overall median for the moderator.

## 3. Discussion

The objective of this protocol is to establish methodological guidelines for a systematic review and meta-analysis to analyse the effects of physical exercise on EI from childhood to adolescence. Among the benefits of developing this process prior to the final review are the careful and documented planning of the methodology to be used (preventing arbitrary decisions and reporting biases) and the rigorous analysis from different perspectives to detect possible errors thanks to peer review, which could favour the replicability of the study and its consequent updating [48].

In turn, the proposed review will approach the described topic from an integrative viewpoint, allowing us to visualize the effects of physical exercise on EI either from its dimensions, its levels, or overall. In this way, it aims to consider the different emotional theories categorized within the EI framework according to the perspective of Mortillaro and Schlegel [5], who propose that other models could also contribute to the classification offered by Salovey and Mayer [2,3,4] (Table 3). This could offer novel information with respect to previous reviews [11,19,20]. In addition, it seeks to understand the possible implications of physical exercise and sports practice as a strategy to improve EI from childhood to adolescence from a broad perspective [47], considering a variety of potential moderators, which could provide greater clarity to the analysis in this area of study.

Some limitations may be encountered during the future conduct of the review. Firstly, it is possible that the number of articles is limited and does not allow for the analysis to be conducted as planned. If this was to occur, the review would be conducted descriptively, calculating the effect size on those variables found. It is also possible that not enough studies will be found for all ages, as in many cases there is a greater amount of research on those over 18 years of age due to the feasibility of the research. Similarly, again, the age groups found will be analysed, indicating not an absence of a relationship but rather a lack of studies. On the other hand, studies may be at risk of bias; however, this will be reported to improve transparency and allow for replicability of the results, analysing their impact on the final conclusions. There may also be problems with the primary data, as not all studies show it in its entirety; however, this will be analysed and considered in the conclusions. Finally, despite all limitations that may be encountered, an attempt will be made to conduct the study as transparently as possible to avoid both of these limitations and the possible subjectivity of the researchers, allowing for replication of the results found.

The results could help in the creation of interdisciplinary interventions in educational or sports centres aimed at promoting EI through strategies that use physical exercise and sport, thus promoting the emotional development of children and young people. The analysis of moderators, meanwhile, will allow for the identification of the relevant characteristics to be considered in interventions based on current evidence, considering relevant aspects of the population such as age and sociocultural variables, which will enrich the application of intervention programs in real contexts.

## Figures and Tables

**Table 1 healthcare-12-02437-t001:** Eligibility criteria.

	Inclusion Criteria	Exclusion Criteria
P	From birth to 21 years.	No gender or status restrictions are contemplated.
I	All acute and chronic interventions that incorporate any type of physical exercise (e.g., endurance; resistance; sport-specific) will be considered. Interventions may or may not include some type of additional EI-related activity.	Interventions that are unsupervised or do not include dosage in their activities (such as exercise prescription or self-paced walking).
C	Control groups not performing the intervention program, either continuing with their usual routines (passive) or performing another activity with very low physical demand (active), and/or additional EI-related activity.	Absence of control group.
O	Outcome measures for EI (as a general index, for each of its dimensions or levels), independently of the theoretical model used.	Measures of EI obtained from assessment with non-validated instruments.
S	Randomized or non-randomized controlled experimental studies.	Studies that do not show the results as posters or congress conferences.

P: population; I: intervention; C: comparison; O: outcome; S: study design.

**Table 2 healthcare-12-02437-t002:** Terms to be used in the search strategy for each database.

Database	Search Strategy
PubMed	((((((((((((((Sport[Title/Abstract]) OR (“Modified sport”[Title/Abstract])) OR (“Physical Exercise”[Title/Abstract])) OR (“Physical Activity”[Title/Abstract])) OR (Athletic*[Title/Abstract])) OR (“resistance training”[Title/Abstract])) OR (swimming[Title/Abstract])) OR (handball[Title/Abstract])) OR (Basketball[Title/Abstract])) OR (“team game”[Title/Abstract])) OR (soccer[Title/Abstract])) OR (Gymnastic*[Title/Abstract])) OR (Volleyball[Title/Abstract])) OR (Tennis[Title/Abstract])) AND (((((((((((Emotion[Title/Abstract]) OR (“Emotional Intelligence”[Title/Abstract])) OR (“Emotion* regulation”[Title/Abstract])) OR (“Emotion* perception”[Title/Abstract])) OR (“Emotion* comprehension”[Title/Abstract])) OR (“Emotion* facilitation”[Title/Abstract])) OR (“social cognition”[Title/Abstract])) OR (“hot executive function*”[Title/Abstract])) OR (“Emotional process*”[Title/Abstract])) OR (“Self-regulation”[Title/Abstract])) OR (“Facial Emotion recognition”[Title/Abstract]))
Web of Science Core Collection	1: AB = (Sport) 2: AB = (“Modified sport”) 3: AB = (“Physical Exercise”) 4: AB = (“Physical Activity”) 5: AB = (Athletic*) 6: AB = (“resistance training”) 7: AB = (swimming) 8: AB = (handball) 9: AB = (Basketball) 10: AB = (“team game”) 11: AB = (soccer) 12: AB = (Gymnastic*) 13: AB = (Volleyball) 14: AB = (Tennis) 15: #1 OR #2 OR #3 OR #4 OR #5 OR #6 OR #7 OR #8 OR #9 OR #10 OR #11 OR #12 OR #13 OR #14 16: AB = (Emotion) 17: AB = (“Emotional Intelligence”) 18: AB = (“Emotion* regulation”) 19: AB = (“Emotion* perception”) 20: AB = (“Emotion* comprehension”) 21: AB = (“Emotion* facilitation”) 22: AB = (“social cognition”) 23: AB = (“hot executive function*”) 24: AB = (“Emotional process*”) 25: AB = (“Self-regulation”) 26: AB = (“Facial Emotion recognition”) 27: #16 OR #17 OR #18 OR #19 OR #20 OR #21 OR #22 OR #23 OR #24 OR #25 OR #26 28: #15 AND #27
Scopus	(ABS (Emotion) OR ABS (“Emotional Intelligence”) OR ABS (“Emotion* regulation”) OR ABS (“Emotion* perception”) OR ABS (“Emotion* comprehension”) OR ABS (“Emotion* facilitation”) OR ABS (“social cognition”) OR ABS (“hot executive function*”) OR ABS (“Emotional process*”) OR ABS (“Self-regulation”) OR ABS (“Facial Emotion recognition”)) AND (ABS (Sport) OR ABS (“Modified sport”) OR ABS (“Physical Exercise”) OR ABS (“Physical Activity”) OR ABS (Athletic*) OR ABS (“resistance training”) OR ABS (swimming) OR ABS (handball) OR ABS (Basketball) OR ABS (“team game”) OR ABS (soccer) OR ABS (Gymnastic*) OR ABS (Volleyball) OR ABS (Tennis))
EBSCO	TX (Sport OR “Modified sport” OR “Physical Exercise” OR “Physical Activity” OR Athletic* OR “resistance training” OR swimming OR handball OR Basketball OR “team game” OR soccer OR Gymnastic* OR Volleyball OR Tennis) AND TX (Emotion OR “Emotional Intelligence” OR “Emotion* regulation” OR “Emotion* perception” OR “Emotion* comprehension” OR “Emotion* facilitation” OR “social cognition” OR “hot executive function*” OR “Emotional process*” OR “Self-regulation” OR “Facial Emotion recognition”)

**Table 3 healthcare-12-02437-t003:** Dimensions of the EI abilities model, other emotional models, and instruments for their evaluation.

Dimensions of EI from the Abilities Model	Other Models	Examples of Evaluation Instruments
Emotional perception	Basic emotion theory, emotion recognition, circumplex model, appraisal theory, theory of constructed emotion, the empathic accuracy paradigm, social cognition	MSCEIT; faces subtest; GERT; EAT; situational judgment approach; WIPS; MASC; Reading the Mind in Films task
Emotional facilitation	This dimension, although not mentioned by the author, evidences a notorious conceptual similarity with hot executive functions [49,50,51]	IGT; delay discounting task; BART; Affective go/no-go task
Emotional understanding	Basic emotion theory, circumplex model, appraisal theory, the empathic agent paradigm, social cognition	MSCEIT; STEU; GECo; GEMOK; CEUT
Emotional regulation (intrapersonal)	Self-regulation, emotion regulation	ERQ; CERQs; PMERQ; GECo; ERP-R
Emotional management (interpersonal)	Conflict management theory, co-enhancing, and co-dampening	MSCEIT; STEM; GECo; IEMS; CoDEQ

Note: Information compiled and summarized from the proposals of Mortillaro and Schlegel [5]. Affective Go/No-Go Task: Tarea afectiva de ir/no ir; BART: Balloon Analogue Risk Task; CEUT: Components of Emotional Understanding Test; CERQs: Cognitive Emotion Regulation Questionnaires; CoDEQ: Co-Dampening and Co-Enhancing Questionnaire; EAT: Emotional Accuracy Test; ERQ: Emotion Regulation Questionnaire; ERP-R: Emotion Regulation Profile Revised; Faces subtest: Face recognition subtest in the MSCEIT; GECo: Geneva Emotional Competence Test; GEMOK: Geneva Emotion Knowledge Test; GERT: Geneva Emotion Recognition Test; IEMS: Interpersonal Emotion Management Scale; IGT: Iowa Gambling Task; MASC: Movie for the Assessment of Social Cognition; MSCEIT: Mayer–Salovey–Caruso Emotional Intelligence Test; PMERQ: Process Model of Emotion Regulation Questionnaire; Reading the Mind in Films Task: task to read the mind through movie scenes; Situational Judgment Approach: situational judgment approach for assessing emotions in contexts; STEM: Situational Test of Emotion Management; STEU: Situational Test of Emotional Understanding; WIPS: Workplace Interpersonal Perception Skill test.

## Data Availability

Data are contained within the article.

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
