# Peer review of "Effects of Physical Exercise on Emotional Intelligence from Birth to Adolescence: A Systematic Review Protocol"

_healthcare, 2024, doi:10.3390/healthcare12232437_

Round 1

Reviewer 1 Report

Comments and Suggestions for Authors

Dear Authors

You intend to conduct a systematic review to investigate the effects of physical exercise on emotional intelligence from birth to adolescence, and then you prepare a protocol for your study. It seems that your protocol is comprehensive enough and well-prepared. I read your protocol in depth and propose some minor recommendations for you. I hope these comments will promote the overall quality of your manuscript. Here you have my comments;

-        Based on the journal guidelines, removing subheadings from your abstract is better.

-        Abstract, please express how you investigate the publication bias in your study.

-        Keyword, I think it is better to include “Emotional Intelligence” and “physical exercise” as your keywords.

-        It is better to conduct a pilot search in one database like PubMed and insert the number of records here. It helps reviewers to inform about the volume of the study included in your search.

-        Line 182, I think it is better to add mean age±SD instead of age because the original studies usually report the mean age or age range of participants.

-        I propose additional items for data extraction, including the country of origin, study design, sampling methods, follow-up time, and funding information. With knowledge of funding information, you can investigate the potential conflict of interest within the included studies. This COI may affect the results of individual studies.

-        What is your solution to deal with heterogeneity in studies? If the heterogeneity in the studies is high, what method will you use to analyze the data?

-        It is better to mention the possible limitations that you will face and the solution to overcome them.

Best regards,

Author Response

Comment: Dear Authors

You intend to conduct a systematic review to investigate the effects of physical exercise on emotional intelligence from birth to adolescence, and then you prepare a protocol for your study. It seems that your protocol is comprehensive enough and well-prepared. I read your protocol in depth and propose some minor recommendations for you. I hope these comments will promote the overall quality of your manuscript. Here you have my comments;

Response: Dear Reviewer,

We sincerely thank you for your time and effort in reviewing our protocol. We greatly appreciate your recommendations, which we view as valuable contributions to enhancing the overall quality of the manuscript.

We are committed to carefully addressing each of your comments to ensure they are reflected in the final version of our research. Thank you again for your support and contribution to this process.

Comment 1: Based on the journal guidelines, removing subheadings from your abstract is better.

Response 1 (lines 12-36): We removed it initially, however, the editor later indicated that the heading should be included and summarised to 250, so we decided to rewrite it.

Comment 2: Abstract, please express how you investigate the publication bias in your study.

Response 2 (line 29): It was added.

Comment 3: Keyword, I think it is better to include “Emotional Intelligence” and “physical exercise” as your keywords.

Response 3 (line 38): Thank you for the suggestion. However, it is generally advised to avoid using title keywords in the abstract or keyword sections, as it can hinder optimal indexing and discoverability in databases. Redundant keywords may reduce visibility and impact. For this reason, we have opted to use alternative keywords to maximize reach and effectiveness. (Pottier et al., 2023; 2024).

References:

Pottier, P., Lagisz, M., Burke, S. G., Drobniak, S. M., Downing, P. A., Macartney, E. L., Martinig, A. R., Mizuno, A., Morrison, K., Pollo, P., Ricolfi, L., Tam, J., Williams, C., Yang, Y., & Nakagawa, S. (2024). Title, abstract and keywords: a practical guide to maximize the visibility and impact of academic papers. Proceedings of The Royal Society B: Biological Sciences, doi: 10.1098/rspb.2024.1222

Pottier, P., Lagisz, M., Burke, S. G., Drobniak, S. M., Downing, P. A., Macartney, E. L., Martinig, A. R., Mizuno, A., Morrison, K., Pollo, P., Ricolfi, L., Tam, J., Williams, C., Yang, Y., & Nakagawa, S. (2023). Keywords to success: a practical guide to maximize the visibility and impact of academic papers. bioRxiv, doi: 10.1101/2023.10.02.559861

Comment 4: It is better to conduct a pilot search in one database like PubMed and insert the number of records here. It helps reviewers to inform about the volume of the study included in your search.

Response 4 (line 201): It was added “A pilot search was conducted in PubMed on October 28, 2024, yielding 2,106 results”.

Comment 5: Line 182, I think it is better to add mean age±SD instead of age because the original studies usually report the mean age or age range of participants.

Response 5 (line 230): it was modified.

Comment 6: I propose additional items for data extraction, including the country of origin, study design, sampling methods, follow-up time, and funding information. With knowledge of funding information, you can investigate the potential conflict of interest within the included studies. This COI may affect the results of individual studies.

Response 6 (line 229): Thank you very much for all the recommendations, we think they are really relevant, and we have added them.

Comment 7: What is your solution to deal with heterogeneity in studies? If the heterogeneity in the studies is high, what method will you use to analyze the data?

Response 7 (line 326): dear reviewer, thank you for your question. To address heterogeneity, the data will be analyzed according to the moderators indicated in the moderators section in order to explore heterogeneity. Accordingly, we will consider the strategies for addressing heterogeneity, proposed by the Cochrane group (Deeks JJ, Higgins JPT, Altman DG (editors). Chapter 9: Analysing data and undertaking meta-analyses. In: Higgins JPT, Green S (editors). Cochrane Handbook for Systematic Reviews of Interventions Version 5.1.0 (updated March 2011). The Cochrane Collaboration, 2011. Available from www.handbook.cochrane.org. Direct link: https://handbook-5-1.cochrane.org/chapter_9/9_5_3_strategies_for_addressing_heterogeneity.htm).

Comment 8: It is better to mention the possible limitations that you will face and the solution to overcome them.

Response 8 (line 375): we think it is relevant so we added: “Some limitations may be encountered during the future conduct of the review. Firstly, it is possible that the number of articles is limited and does not allow the analyses to be carried out as planned. If this were to occur, the review would be conducted descriptively, calculating the effect size on those variables found. It is also possible that not enough studies will be found for all ages, as in many cases there is a greater amount of research on those over 18 years of age due to the feasibility of the research. Similarly, again, the age groups found will be analysed, indicating not the absence of a relationship but the lack of studies. On the other hand, studies may be at risk of bias, however, this will be reported to improve transparency and allow for replicability of the results, analysing their possible impact on the final conclusions. There may also be problems with the primary data, as not all studies show it in its entirety, however, this will be analysed and considered in the conclusions. Finally, despite all possible limitations that may be encountered, an attempt will be made to conduct the study as transparently as possible to avoid both these limitations and the possible subjectivity of the researchers, allowing for replication of the results found”.

Best regards,

Reviewer 2 Report

Comments and Suggestions for Authors

 It is suggested to remove the word "protocol" in the title.

What is meant by the period from birth to adolescence in the title? In the abstract you said until 21 years? 21 years old is considered a young adult but not late adolescence

Indicate in the abstract that the search was carried out exactly in what historical period? And what period of time have you reviewed the articles?

line 21. Does "two independent reviewers" refer to some of the authors of this manuscript?

 Although the authors tried to mention the necessity of conducting this research in the introduction, it seems that there is still a need to better address this issue at the end of the introduction. Why exactly should this research be done? Try to explain a little more.

 In the results section, it seems that the results are not well mentioned? It is suggested that the results be presented in some tables and that the studies that met all the inclusion criteria and were properly reported should be described with full explanations.

 The discussion seems to be poorly written, for example, the limitations of the research are not well mentioned. The application of the results should be better mentioned.

Author Response

Dear Reviewer,

We are very grateful for your thoughtful review and for the time you dedicated to providing feedback on our protocol. Your insights and suggestions are invaluable, and we believe they will significantly strengthen our work.

We will carefully consider and integrate your comments to enhance the manuscript further. Thank you again for your constructive feedback and support in refining our research.

Comment 1: It is suggested to remove the word "protocol" in the title.

Response 1 (line 1): We appreciate the comment, however we believe it is necessary as the PRISMA guideline for protocols states the following in item 1a: Identify the report as a protocol of a systematic review.

Moher, D., Shamseer, L., Clarke, M., Ghersi, D., Liberati, A., Petticrew, M., ... & Prisma-P Group. (2015). Preferred reporting items for systematic review and meta-analysis protocols (PRISMA-P) 2015 statement. Systematic reviews, 4, 1-9.

Comment 2: What is meant by the period from birth to adolescence in the title? In the abstract you said until 21 years? 21 years old is considered a young adult but not late adolescence

Response 2: It is true that other classifications indicate lower ages, however, we chose the UNICEF classification of adolescence, which indicates that late adolescence goes up to the age of 21.

United Nations International Children’s Emergency Fund (UNICEF) ¿Qué Es La Adolescencia? Available online: https://www.unicef.org/uruguay/crianza/adolescencia/que-es-la-adolescencia (accessed on 3 October 2024).

Comment 3: Indicate in the abstract that the search was carried out exactly in what historical period? And what period of time have you reviewed the articles?

Response 3 (lines 12-36): The review has not yet taken place as it is a protocol, so there is no date yet. However, this date will be set right after the acceptance of the protocol. This information was added in the abstract.

Comment 4: line 21. Does "two independent reviewers" refer to some of the authors of this manuscript?

Response 4: yes, this is explained in the methodology section together with the authors' acronyms, however, we do not think it is necessary to add it in the abstract.

Comment 5: Although the authors tried to mention the necessity of conducting this research in the introduction, it seems that there is still a need to better address this issue at the end of the introduction. Why exactly should this research be done? Try to explain a little more.

Response 5 (line 162): thank you for the comment, it was added: “Despite evidence on the psychological benefits of exercise, the relationship between physical activity and emotional intelligence (EI) remains underexplored. Given that EI is key to stress management and social interactions, understanding how exercise influences EI is individually and socially relevant. This research could provide a basis for interventions that strengthen EI through exercise, benefiting health professionals and the wider community”.

Comment 6: In the results section, it seems that the results are not well mentioned? It is suggested that the results be presented in some tables and that the studies that met all the inclusion criteria and were properly reported should be described with full explanations.

Response 6: As this study is a protocol, i.e. the document describing the work plan, the results have not yet been obtained. The protocol allows the methodology and design of the study to be reviewed by peer reviewers to improve and check that the study is relevant and properly designed. In this way, it improves the future review that will be executed after the publication of the protocol. Therefore, there are no results to show.

Comment 7: The discussion seems to be poorly written, for example, the limitations of the research are not well mentioned. The application of the results should be better mentioned.

Response 7: thank you for the comment, we modified and added according to your comments.

Reviewer 3 Report

Comments and Suggestions for Authors

Dear Author;

I think the research design is commendable and promises to make a significant contribution to the literature. I look forward to your study results. I recommend some revisions;

The introduction is too long. you should write the literature information as an introduction, not as a discussion

spelling errors and English should be revised (for example; Page 1 line 12. “emotional” should be written capital letter.)

References should be revised according to the guidelines.

Comments on the Quality of English Language

Should be revised by paying attention to the concepts of tense and tense sentences.

Author Response

Comment: Dear Author;

I think the research design is commendable and promises to make a significant contribution to the literature. I look forward to your study results. I recommend some revisions;

Response: Dear Reviewer,

Thank you very much for your encouraging feedback on our research design for the protocol. We appreciate your recognition of our study's potential to contribute meaningfully to the literature, and we are equally eager to share the results.

We will carefully review and incorporate your recommended revisions to further improve the quality of our work. Thank you once again for your valuable insights.

Comment 1: The introduction is too long. you should write the literature information as an introduction, not as a discussion

Response 1: The introduction was shortened to make it more concise, please let us know if you think it needs to be reduced further.

Comment 2: spelling errors and English should be revised (for example; Page 1 line 12. “emotional” should be written capital letter.)

Response 2: It was reviewed.

Comment 3: References should be revised according to the guidelines.

Response 3: we use mendeley for the references and use the journal's referencing style, we have also double-checked them all.

Comment 4: Comments on the Quality of English Language

Should be revised by paying attention to the concepts of tense and tense sentences.

Response 4: The full text was revised and some terms were changed to improve the wording, although it should be noted that the entire text is in the future tense because it is a protocol.

Reviewer 4 Report

Comments and Suggestions for Authors

First of all, I would like to congratulate the authors for the manuscript submitted to this journal. The work answers the relevant questions and is of sufficient quality to be published.

Summary and keywords

Clear, concise and very complete.

Introduction

Complete introduction. The variables that will later be taken into account in the review are discussed.

The significance statistic is in italics. For example “p”.

Method

It is presented clearly and concisely.

On line 141 a punctuation mark has been inserted that does not make sense with the sentence.

In the point of Selection Process (line 165), the search flow typical of systematic reviews should appear. 

Line 278. If it has been previously decided that the significance statistic should have a 0 in front, it should always be put that way. In other words, the way in which the values appear must be homogenized.

Results

A results section is missing, practically all results are presented in the method. The method section should explain what has been done and how. However, a results section is necessary to present the findings obtained.

A summary table including all the studies included in the systematic review should be presented. This table should reflect all the data presented in the results.

Discussion

Discussion too short. The main results should be extracted and compared with studies that agree and disagree. Additionally, the causes and consequences of why this might occur should be added. 

Homogenize the way in which citations are put. For example, in line 311 add three citations and put the three numbers and in table 3 put three citations with a hyphen in the middle and omitting the central citation. Choose one way or the other.

Conclusions

There is no conclusion section, which is of great importance to conclude an article. Additionally, strengths and limitations should also be shown in this section. A new section should be created to clearly state the latter.

Bibliographic references

Put the doi links

Check that the citations correspond to the number of the references.

Author Response

Response: We would like to express our sincere gratitude to the reviewer for their thoughtful evaluation and kind words. We appreciate the positive feedback and are pleased that our work has been recognized as addressing relevant questions and meeting the quality standards for publication.

Comment: First of all, I would like to congratulate the authors for the manuscript submitted to this journal. The work answers the relevant questions and is of sufficient quality to be published.

Comment 1: Summary and keywords. Clear, concise and very complete.

Response 1: Thank you for your positive feedback

Comment 2: Introduction: Complete introduction. The variables that will later be taken into account in the review are discussed. The significance statistic is in italics. For example “p”.

Response 2: Thank you for your feedback on the introduction. We have reviewed and revised the statistical notations, ensuring that the significance statistics, such as "p," are properly formatted in italics as suggested.

Comment 3: Method: It is presented clearly and concisely. On line 141 a punctuation mark has been inserted that does not make sense with the sentence.

Response 3: we have reviewed the sentence on line 141 and removed the punctuation mark that was incorrectly placed, ensuring the sentence flows correctly and maintains clarity.

Comment 4: In the point of Selection Process (line 165), the search flow typical of systematic reviews should appear.

Response 4:  Thank you for your suggestion. We have added an indication that final data will be collected and presented in the completed review. However, we would like to clarify that this manuscript is a systematic review and meta-analysis protocol, and therefore, a detailed search flow diagram typically included in systematic reviews is not required here. Since the purpose of a protocol is to outline the planned methodology rather than report findings, the search flow diagram will be included in the final review once data collection and analysis are complete.

Comment 5: Line 278. If it has been previously decided that the significance statistic should have a 0 in front, it should always be put that way. In other words, the way in which the values appear must be homogenized.

Response 5: Thank you for your observation. We have reviewed the manuscript and ensured consistency in the presentation of statistical significance values by adding a zero in front.

Comment 6: Results. A results section is missing, practically all results are presented in the method. The method section should explain what has been done and how. However, a results section is necessary to present the findings obtained. A summary table including all the studies included in the systematic review should be presented. This table should reflect all the data presented in the results.

Response 6: Thank you for your comments. However, we would like to clarify that our manuscript is a systematic review and meta-analysis protocol. In a protocol, results are not included, as the purpose of this publication is to detail the plan and methods that will be used to conduct the review and meta-analysis. Including a results section or a summary table of studies is not appropriate at this stage, as data have not yet been collected or analyzed. This structure is standard in protocols and aligns with international guidelines for systematic review protocols.

Comment 7: Discussion too short. The main results should be extracted and compared with studies that agree and disagree. Additionally, the causes and consequences of why this might occur should be added. 

Response 7: Thank you for your comment. However, we would like to clarify that, as this is a systematic review and meta-analysis protocol, no results are yet available to discuss or compare with existing studies. The purpose of a protocol is to outline the planned methodology rather than to present findings. Therefore, an extended discussion analyzing results is not applicable at this stage. The discussion section in the final review will address these points once data have been collected and analyzed.

Comment 8: Homogenize the way in which citations are put. For example, in line 311 add three citations and put the three numbers and in table 3 put three citations with a hyphen in the middle and omitting the central citation. Choose one way or the other.

Response 8: Thank you for your comment. We would like to clarify that the citation format varies intentionally to reflect the inclusion or exclusion of non-consecutive references. In line 311, we used commas between numbers (13, 21, 22) to indicate non-sequential citations, meaning references 14, 15, etc., are not included. Conversely, in instances where we use a hyphen (e.g., 2-4), it indicates a continuous sequence, inclusive of all intervening references (in this case, including 3).

Comment 9: Conclusions. There is no conclusion section, which is of great importance to conclude an article. Additionally, strengths and limitations should also be shown in this section. A new section should be created to clearly state the latter.

Response 9: Thank you for your comment. As this manuscript is a systematic review and meta-analysis protocol, a conclusion section is not typically included, as the purpose of a protocol is to outline the planned methodology rather than present findings or conclusions. However, in response to this and other reviewers' suggestions, we have added a section discussing potential limitations that may arise during the study.

Comment 10: Bibliographic references. Put the doi links. Check that the citations correspond to the number of the references.

Response 10: Thank you for your observation. We have carefully reviewed all bibliographic references and ensured that citations correspond accurately to their reference numbers. Additionally, we use the Mendeley reference manager to minimize human error in managing citations. If you notice any specific discrepancies, please let us know, and we will address them promptly.

Reviewer 5 Report

Comments and Suggestions for Authors

The article proposes a systematic review to examine the effects of physical exercise on emotional intelligence (EI) in individuals from birth through adolescence following the PRISMA-P guidelines and registered in PROSPERO.

There are several aspects to be taken into account in order to improve the manuscript:

In the abstract and in the line 126, it is stated that the aim of the manuscript is “To systematically review studies that examined the acute and chronic effects of physical exercise on EI in participants aged from birth to 21 years (late adolescence).” and "Therefore, the aim of this review is to analyze the effects of physical exercise on the different dimensions of EI - perception, facilitation, knowledge, regulation, and emotional management - in individuals from birth to adolescence.

However, what has been done is a protocol proposal, so the objective cannot be that.

In this sense, I understand that the objective should be reformulated or the proposal made by the authors should be carried out, thus carrying out the systematic review.

In addition, in the Inclusion and Exclusion Criteria it is indicated in the inclusion criteria that it is up to 21 years of age, but in PROSPERO, in the inclusion criteria, it appears up to 24 years of age. This aspect should be corrected.

On the other hand, the following minor bugs should be corrected:

Correct line 41: “y Salovey”.

Correct line 317: Change the word “nota” to “note”.

Author Response

Comment: The article proposes a systematic review to examine the effects of physical exercise on emotional intelligence (EI) in individuals from birth through adolescence following the PRISMA-P guidelines and registered in PROSPERO.

There are several aspects to be taken into account in order to improve the manuscript:

Response: Dear Reviewer,

Thank you for your thorough review and thoughtful feedback on our manuscript. We are grateful for your careful attention to detail and your suggestions for improvement, which will undoubtedly help us refine our study.

We will address each of your recommendations in line with the PRISMA-P guidelines and make the necessary adjustments to strengthen the manuscript further. Thank you again for your valuable insights and support.

Comment 1: In the abstract and in the line 126, it is stated that the aim of the manuscript is “To systematically review studies that examined the acute and chronic effects of physical exercise on EI in participants aged from birth to 21 years (late adolescence).” and "Therefore, the aim of this review is to analyze the effects of physical exercise on the different dimensions of EI - perception, facilitation, knowledge, regulation, and emotional management - in individuals from birth to adolescence. However, what has been done is a protocol proposal, so the objective cannot be that. In this sense, I understand that the objective should be reformulated or the proposal made by the authors should be carried out, thus carrying out the systematic review.

Response 1: it was changed by “the aim of this protocol was to establish methodological guidelines for a systematic review and meta-analysis of the effects of acute and chronic physical exercise on EI - perception, facilitation, knowledge, regulation, and emotional management - from birth to 21 years (late adolescence)” in the text and in the abstract.

Comment 2: In addition, in the Inclusion and Exclusion Criteria it is indicated in the inclusion criteria that it is up to 21 years of age, but in PROSPERO, in the inclusion criteria, it appears up to 24 years of age. This aspect should be corrected.

Response 2: Thank you for the thorough review. Indeed, this protocol was previously registered in PROSPERO, indicating that age at the beginning. However, as mentioned in the methodology, this record will be modified after peer review once the paper is published. This is because we believe that peer review is beneficial for the improvement of the article and is the main motivation for publication of the protocol.

Comment 3: On the other hand, the following minor bugs should be corrected: Correct line 41: “y Salovey”. Correct line 317: Change the word “nota” to “note”.

Response 3: it was modified.

Round 2

Reviewer 5 Report

Comments and Suggestions for Authors

Dear authors,

After re-reading the paper, I feel that it has been greatly improved. I see that you have taken into consideration the comments of the first revision and made other changes, improving the work considerably.

I have not found any aspects that require improvement, so I consider that the work is suitable for publication.

Congratulations for the work done.